# Representation Degeneration Problem in Training Natural Language Generation Models

**Jun Gao**[1,2*]**, Di He**[3]**, Xu Tan**[4]**, Tao Qin**[4]**, Liwei Wang**[3,5] **& Tie-Yan Liu**[4]

[1]Department of Computer Science, University of Toronto

`jungao@cs.toronto.edu`

[2]Vector Institute, Canada

[3]Key Laboratory of Machine Perception, MOE, School of EECS, Peking University

`di_he@pku.edu.cn, wanglw@cis.pku.edu.cn`

[4]Microsoft Research

`{xuta,taoqin,tyliu}@microsoft.com`

[5]Center for Data Science, Peking University, Beijing Institute of Big Data Research

## Abstract

We study an interesting problem in training neural network-based models for natural language generation tasks, which we call the *representation degeneration problem*. We observe that when training a model for natural language generation tasks through likelihood maximization with the weight tying trick, especially with big training datasets, most of the learnt word embeddings tend to degenerate and be distributed into a narrow cone, which largely limits the representation power of word embeddings. We analyze the conditions and causes of this problem and propose a novel regularization method to address it. Experiments on language modeling and machine translation show that our method can largely mitigate the representation degeneration problem and achieve better performance than baseline algorithms.

## 1 Introduction

Neural Network (NN)-based algorithms have made significant progresses in natural language generation tasks, including language modeling (Kim et al., 2016; Jozefowicz et al., 2016), machine translation (Wu et al., 2016; Britz et al., 2017; Vaswani et al., 2017; Gehring et al., 2017) and dialog systems (Shang et al., 2015). Despite the huge variety of applications and model architectures, natural language generation mostly relies on predicting the next word given previous contexts and other conditional information. A standard approach is to use a deep neural network to encode the inputs into a fixed-size vector referred as *hidden state*[1], which is then multiplied by the word embedding matrix (Vaswani et al., 2017; Merity et al., 2018; Yang et al., 2018; Press & Wolf, 2017). The output logits are further consumed by the softmax function to give a categorical distribution of the next word. Then the model is trained through likelihood maximization.

While studying the learnt models for language generation tasks, we observe some interesting and surprising phenomena. As the word embedding matrix is tied with the softmax parameters (Vaswani et al., 2017; Merity et al., 2018; Inan et al., 2017; Press & Wolf, 2017), it has a *dual role* in the model, serving as the input in the first layer and the weights in the last layer. Given its first role as input word embedding, it should contain rich semantic information that captures the given context which will be further used for different tasks. Given its role as output softmax matrix, it should have enough capacity to classify different hidden states into correct labels. We compare it with

---

[*]This work was done when Jun Gao was an intern at Microsoft Research Asia.

[1]The concept of hidden states has multiple meanings in the literature of neural networks. In this paper, we use *hidden state* as the input to the last softmax layer.

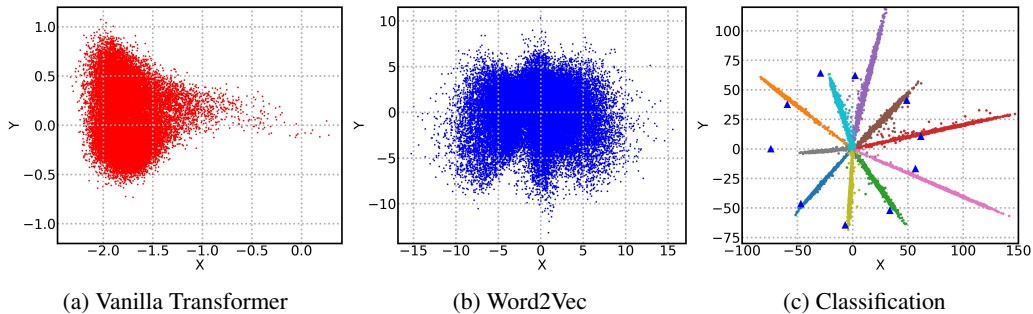

|       |       |       |
|-------|-------|-------|
| (a) Vanilla Transformer | (b) Word2Vec | (c) Classification |

Figure 1: 2D visualization. (a). Visualization of word embeddings trained from vanilla Transformer (Vaswani et al., 2017) in English→German translation task. (b). Visualization of word embeddings trained from Word2Vec (Mikolov et al., 2013). (c). Visualization of hidden states and category embedding of a classification task, where different colors stand for different categories and the blue triangles denote for category embeddings.

word embeddings trained from Word2Vec (Mikolov et al., 2013) and the parameters in the softmax layer of a classical classification task (we refer it as categorical embedding). As shown in Figure 1, the word embeddings learnt from Word2Vec (Figure 1(b)) and the softmax parameters learnt from the classification task (Figure 1(c)) are diversely distributed around the origin using SVD projection; in contrast, the word embeddings in our studied model (Figure 1(a)) degenerated into a narrow cone. Furthermore, we find the embeddings of any two words in our studied models are positively correlated. Such phenomena are very different from those in other tasks and deteriorate model's capacity. As the role of the softmax layer, those parameters cannot lead to a large margin prediction for good generalization. As the role of word embeddings, the parameters do not have enough capacity to model the diverse semantics in natural languages (Yang et al., 2018; McCann et al., 2017).

We call the problem described above the *representation degeneration problem*. In this paper, we try to understand why the problem happens and propose a practical solution to address it.

We provide some intuitive explanation and theoretical justification for the problem. Intuitively speaking, during the training process of a model with likelihood loss, for any given hidden state, the embedding of the corresponding ground-truth word will be pushed towards the direction of the hidden state in order to get a larger likelihood, while the embeddings of all other words will be pushed towards the negative direction of the hidden state to get a smaller likelihood. As in natural language, word frequency is usually very low comparing to the size of a large corpus, the embedding of the word will be pushed towards the negative directions of most hidden states which drastically vary. As a result, the embeddings of most words in the vocabulary will be pushed towards similar directions negatively correlated with most hidden states and thus are clustered together in a local region of the embedding space.

From the theoretical perspective, we first analyze the extreme case of non-appeared words. We prove that the representation degeneration problem is related to the structure of hidden states: the degeneration appears when the convex hull of the hidden states does not contain the origin and such condition is likely to happen when training with *layer normalization* (Ba et al., 2016; Vaswani et al., 2017; Merity et al., 2018). We further extend our study to the optimization of low-frequency words in a more realistic setting. We show that, under mild conditions, the low-frequency words are likely to be trained to be close to each other during optimization, and thus lie in a local region.

Inspired by the empirical analysis and theoretical insights, we design a novel way to mitigate the degeneration problem by regularizing the word embedding matrix. As we observe that the word embeddings are restricted into a narrow cone, we try to directly increase the size of the aperture of the cone, which can be simply achieved by decreasing the similarity between individual word embeddings. We test our method on two tasks, language modeling and machine translation. Experimental results show that the representation degeneration problem is mitigated, and our algorithm achieves superior performance over the baseline algorithms, e.g., with 2.0 point perplexity improvement on

the WikiText-2 dataset for language modeling and 1.08/0.93 point BLEU improvement on WMT 2014 English-German/German-English tasks for machine translation.

## 2  RELATED WORK

Language modeling and machine translation are important language generation tasks. Language modeling aims at predicting the next token given an (incomplete) sequence of words as context. A popular approach based on neural networks is to map the given context to a real-valued vector as a hidden state, and then pass the hidden state through a softmax layer to generate a distribution over all the candidate words. There are different choices of neural networks used in language modeling. Recurrent neural network-based model and convolutional neural network-based model are widely used (Mikolov et al., 2010; Dauphin et al., 2017).

Neural Machine Translation (NMT) is a challenging task that has attracted lots of attention in recent years. Based on the encoder-decoder framework, NMT starts to show promising results in many language pairs. The evolving structures of NMT models in recent years have pushed the accuracy of NMT into a new level. The attention mechanism (Bahdanau et al., 2015) added on top of the encoder-decoder framework is shown to be useful to automatically find alignment structure, and single-layer RNN-based structure has evolved towards deep models (Wu et al., 2016), more efficient CNN models (Gehring et al., 2017), and well-designed self-attention models (Vaswani et al., 2017).

In this paper, we mainly study the expressiveness of word embeddings in language generation tasks. A trend for language generation in recent years is to share the parameters between word embeddings and the softmax layer, which is named as the weight tying trick. Many state-of-the-art results in language modeling and machine translation are achieved with this trick (Vaswani et al., 2017; Merity et al., 2018; Yang et al., 2018; Press & Wolf, 2017). Inan et al. (2017) shows that weight tying not only reduces the number of parameters but also has theoretical benefits.

## 3  REPRESENTATION DEGENERATION PROBLEM

In this section, we empirically study the word embeddings learnt from sequence generation tasks, i.e., machine translation, and introduce the *representation degeneration problem* in neural sequence generation.

### 3.1  EXPERIMENTAL DESIGN

Our analysis reported in this section is mainly based on the state-of-the-art machine translation model Transformer (Vaswani et al., 2017). We use the official code (Vaswani et al., 2018) and set all the hyperparameters (the configurations of the `base` Transformer) as default. We train the model on the WMT 2014 English-German Dataset and achieve 27.3 in terms of BLEU score. Additionally, we also analyze the LSTM-based model (Wu et al., 2016) and find the observations are similar.

In neural sequence generation tasks, the weights in word embeddings and softmax layer are tied. Those parameters can be recognized not only as the inputs in the first layer but also as the weights in the last layer. Thus we compare this weight matrix with a word embedding matrix trained from conventional word representation learning task, and also compare it with the parameters in softmax layer of a conventional classification task. For simplicity and representative, we choose to use Word2Vec (Mikolov et al., 2013) to obtain word embeddings and use MNIST as the classification task trained with a two-layer convolutional neural network (Yue et al., 2018; Liu et al., 2016). For the classification task, We simply treat the row of the parameter matrix in the last softmax layer as the embedding for the category, like how the word embedding is used in the softmax layer for neural language generation model with weight tying trick.

To get a comprehensive understanding, we use low-rank approximation ($rank = 2$) for the learned matrices by SVD and plot them in a 2D plane, as shown in Figure 1. We also check the singular values distribution and find that our low-rank approximation is reasonable: other singular values are much smaller than the chosen ones.

## 3.2 DISCUSSION

In the classification task, the category embeddings (blue triangles in Figure 1(c)) in the softmax layer are diversely distributed around the origin. This shows the directions of category embeddings are different from each other and well separated, which consequently leads to large margin classification results with good generalization. For the word embeddings learnt from Word2Vec (Figure 1(b)), the phenomena are similar, the embeddings are also widely distributed around the origin in the projection space, which shows different words have different semantic meanings. Observations on GLOVE (Pennington et al., 2014) reported in Mu et al. (2018) are also consistent with ours.

We observe very different phenomena for machine translation. We can see from Figure 1(a) that the word embeddings are clustered together and only lying in a narrow cone. Furthermore, we find the cosine similarities between word embeddings are positive for almost all cases. That is, the words huddle together and are not well separated in the embedding space.

Clearly, as the role of word embeddings, which are the inputs to the neural networks, the word representations should be widely distributed to represent different semantic meanings. As the role of softmax in the output layer, to achieve good prediction of next word in a target sentence, a more diverse distribution of word embeddings in the space is expected to obtain a large margin result. However, such representations of words limit the expressiveness of the learnt model. We call such a problem the *representation degeneration problem*.

## 4 UNDERSTANDING THE PROBLEM

We show in the previous section that in training natural language generation models, the learnt word embeddings are clustered into a narrow cone and the model faces the challenge of limited expressiveness. In this section, we try to understand the reason of the problem and show that it is related to the optimization of low-frequency words with diverse contexts.

In natural language generation tasks, the vocabulary is usually of large size and the words are of low frequencies according to Zipf's law. For example, more than 90% of words' frequencies are lower than 10e-4 in WMT 2014 English-German dataset. Note that even for a popular word, its frequency is also relatively low. For a concrete example, the frequency of the word "is" is only about 1% in the dataset, since "is" occurs at most once in most simple sentences. Our analysis is mainly based on the optimization pattern of the low-frequency words which occupy the major proportion of vocabulary.

The generation of a sequence of words (or word indexes) $y = (y_1, \cdots, y_M)$ is equivalent to generate the words one by one from left to right. The probability of generating $y$ can be factorized as $P(Y = y) = \Pi_t P(Y_t = y_t | Y_{<t} = y_{<t})$, where $y_{<t}$ denotes for the first $t - 1$ words in $y$. Sometimes, the generative model also depends on other context, e.g., the context from the source sentence for machine translation. To study the optimization of the word embeddings, we simply consider the generation of a word as a multi-class classification problem and formally describe the optimization as follows.

Consider a multi-class classification problem with $M$ samples. Let $h_i$ denote the hidden state before the softmax layer which can be also considered as the input features, $i = 1, \cdots, M$. Without any loss of generality, we assume $h_i$ is not a zero vector for all $i$. Let $N$ denote the vocabulary size. The conditional probability of $y_i \in \{1, \cdots, N\}$ is calculated by the softmax function: $P(Y_i = y_i | h_i) = \frac{\exp(\langle h_i, w_{y_i} \rangle)}{\sum_{l=1}^{N} \exp(\langle h_i, w_l \rangle)}$, where $w_l$ is the embedding for word/category $l$, $l = 1, 2, \cdots, N$.

### 4.1 EXTREME CASE: NON-APPEARED WORD TOKENS

Note that in most NLP tasks, the frequencies of rare words are rather low, while the number of them is relatively large. With stochastic training paradigm, the probabilities to sample a certain infrequent word in a mini-batch are very low, and thus during optimization, the rare words behave similarly to a non-appeared word. We first consider the extreme case of non-appeared word in this section and extend to a more general and realistic scenario in the next section.

We assume $y_i \neq N$ for all $i$. That is, the $N$-th word with embedding $w_N$ does not appear in the corpus, which is the extreme case of a low-frequency rare word. We focus on the optimization

process of $w_N$ and assume all other parameters are fixed and well-optimized. By log-likelihood maximization, we have

$$\max_{w_N} \frac{1}{M} \sum_{i=1}^{M} \log \frac{\exp(\langle h_i, w_{y_i} \rangle)}{\sum_{l=1}^{N} \exp(\langle h_i, w_l \rangle)}. \tag{1}$$

As all other parameters are fixed, this is equivalent to

$$\min_{w_N} \frac{1}{M} \sum_{i=1}^{M} \log(\exp(\langle h_i, w_N \rangle) + C_i), \tag{2}$$

where $C_i = \sum_{l=1}^{N-1} \exp(\langle h_i, w_l \rangle)$ and can be considered as some constants.

**Definition 1.** *We say that vector $v$ is a uniformly negative direction of $h_i$, $i = 1, \cdots, M$, if $\langle v, h_i \rangle < 0$ for all $i$.*

The following theorem provides a sufficient condition for the embedding $w_N$ approaching unbounded during optimization. We leave the proof of all the theorems in the appendix.

**Theorem 1.** *A. If the set of uniformly negative direction is not empty, it is convex. B. If there exists a $v$ that is a uniformly negative direction of $h_i$, $i = 1, \cdots, M$, then the optimal solution of Eqn. 2 satisfies $\| w_N^* \| = \infty$ and can be achieved by setting $w_N^* = \lim_{k \to +\infty} k \cdot v$.*

From the above theorem, we can see that if there exists a set of uniformly negative direction, the embedding $w_N$ can be optimized along any uniformly negative direction to infinity. As the set of uniformly negative direction is convex, $w_N$ is likely to lie in a convex cone and move to infinity during optimization. Next, we provide a sufficient and necessary condition for the existence of the uniformly negative direction.

**Theorem 2.** *There exists a $v$ that is a uniformly negative direction of a set of hidden states, if and only if the convex hull of the hidden states does not contain the origin.*

**Discussion on whether the condition happens in real practice** From the theorem, we can see that the existence of the uniformly negative direction is highly related to the structure of the hidden states, and then affects the optimization of word embeddings. Note that a common trick used in sequence generation tasks is layer normalization (Ba et al., 2016; Vaswani et al., 2017; Merity et al., 2018), which first normalizes each hidden state vector into standard vector[2], and then rescales/translates the standard vector with scaling/bias term. In the appendix, we show that under very mild conditions, the space of hidden states doesn't contain the origin almost for sure in practice.

## 4.2 EXTENSION TO RARELY APPEARED WORD TOKENS

In the previous section, we show that under reasonable assumptions, the embeddings of all non-appeared word tokens will move together along the uniformly negative directions to infinity. However, it is not realistic that there exist non-appeared word tokens and the weights of embedding can be trained to be unbounded with L2 regularization term. In this section, we extend our analysis to a more realistic setting. The key we want to show is that the optimization of a rarely appeared word token is similar to that of non-appeared word tokens. Following the notations in the previous section, we denote the embedding of a rare word as $w_N$ and fix all other parameters. We study the optimization for this particular token with the negative log-likelihood loss function.

To clearly characterize the influence of $w_N$ to the loss function, we divide the loss function into two pieces. Piece $A_{w_N}$ contains the sentences that do not contain $w_N$, i.e., all hidden states in such sentences are independently of $w_N$ and the ground truth label of each hidden state (denoted as $w_h^*$) is not $w_N$. Then $w_N$ can be considered as a "non-appeared token" in this set. Denote the space of the hidden state in piece $A_{w_N}$ as $\mathcal{H}_{A_{w_N}}$ with probability distribution $P_{A_{w_N}}$, where $\mathcal{H}_{A_{w_N}}$ and $P_{A_{w_N}}$ can be continuous. The loss function on piece $A_{w_N}$ can be defined as

$$L_{A_{w_N}}(w_N) = - \int_{\mathcal{H}_{A_{w_N}}} \log \frac{\exp(\langle h, w_h^* \rangle)}{\sum_{l=1}^{N} \exp(\langle h, w_l \rangle)} dP_{A_{w_N}}(h), \tag{3}$$

---

[2]the mean and variance of the values of the vector are normalized to be 0 and 1 respectively

which is a generalized version of Eqn. 1. Piece $B_{w_N}$ contains the sentences which contain $w_N$, i.e., $w_N$ appears in every sentence in piece $B_{w_N}$. Then in piece $B_{w_N}$, in some cases the hidden states are computed based on $w_N$, e.g., when the hidden state is used to predict the next token after $w_N$. In some other cases, the hidden states are used to predict $w_N$. Denote the space of the hidden state in piece $B_{w_N}$ as $\mathcal{H}_{B_{w_N}}$ with probability distribution $P_{B_{w_N}}$. The loss function on piece $B_{w_N}$ can be defined as

$$L_{B_{w_N}}(w_N) = -\int_{\mathcal{H}_{B_{w_N}}} \log \frac{\exp(\langle h, w_h^* \rangle)}{\sum_{l=1}^N \exp(\langle h, w_l \rangle)} dP_{B_{w_N}}(h), \tag{4}$$

Based on these notations. the overall loss function is defined as

$$L(w_N) = P(\text{sentence } s \text{ in } A_{w_N}) L_{A_{w_N}}(w_N) + P(\text{sentence } s \text{ in } B_{w_N}) L_{B_{w_N}}(w_N). \tag{5}$$

The loss on piece $A_{w_N}$ is convex while the loss on piece $B_{w_N}$ is complicated and usually non-convex with respect to $w_N$. The general idea is that given the fact that $L_{A_{w_N}}(w_N)$ is a convex function with nice theoretical properties, if $P(\text{sentence } s \text{ in } A_{w_N})$ is large enough, e.g., larger than $1 - \epsilon$, and $L_{B_{w_N}}(w_N)$ is a bounded-smooth function. The optimal solution of $L(w_N)$ is close to the optimal solution of $L_{A_{w_N}}(w_N)$ which can be unique if $L_{A_{w_N}}(w_N)$ is strongly-convex. A formal description is as below.

**Theorem 3.** *Given an $\alpha$-strongly convex function $f(x)$ and a function $g(x)$ that satisfies its Hessian matrix $\mathbf{H}(g(x)) \succ -\beta I$, where $I$ is the identity matrix, and $|g(x)| < B$. For a given $\epsilon > 0$, let $x^*$ and $x_\epsilon^*$ be the optimum of $f(x)$ and $(1 - \epsilon)f(x) + \epsilon g(x)$, respectively. If $\epsilon < \frac{\alpha}{\alpha + \beta}$, then $\| x^* - x_\epsilon^* \|_2^2 \le \frac{4\epsilon B}{\alpha - \epsilon(\alpha + \beta)}$.*

We make some further discussions regarding the theoretical results. In natural language generation, for two low-frequency words $w$ and $w'$, piece $A_w$ and $A_{w'}$ has large overlaps. Then the loss $L_{A_w}(w)$ and $L_{A_{w'}}(w')$ are similar with close optimum. According to the discussion above, the learnt word embedding of $w$ and $w'$ is likely to be close to each other, which is also observed from the empirical studies.

## 5 ADDRESSING THE PROBLEM

In this section, we propose an algorithm to address the representation degeneration problem. As shown in the previous study, the learnt word embeddings are distributed in a narrow cone in the Euclid space which restricts the expressiveness of the representation. Then a straightforward approach is to improve the aperture of the cone which is defined as the maximum angle between any two boundaries of the cone. For the ease of optimization, we minimize the cosine similarities between any two word embeddings to increase the expressiveness.

For simplicity, we denote the normalized direction of $w$ as $\hat{w}$, $\hat{w} = \frac{w}{\|w\|}$. Then our goal is to minimize $\sum_i \sum_{j \neq i} \hat{w}_i^T \hat{w}_j$ as well as the original log-likelihood loss function. By introducing hyperparameter $\gamma$ to trade off the log-likelihood loss and regularization term, the overall objective is,

$$L = L_{\text{MLE}} + \gamma \frac{1}{N^2} \sum_i^N \sum_{j \neq i}^N \hat{w}_i^T \hat{w}_j. \tag{6}$$

We call this new loss as MLE with Cosine Regularization (MLE-CosReg). In the following, we make some analysis about the regularization term.

Denote $R = \sum_i^N \sum_{j \neq i}^N \hat{w}_i^T \hat{w}_j$, and denote the normalized word embedding matrix as $\hat{W} = [\hat{w}_1, \hat{w}_2, ..., \hat{w}_N]^T$. It is readily to check the regularizer has the matrix form as $R = \sum_i^N \sum_{j \neq i}^N \hat{w}_i^T \hat{w}_j = \sum_i^N \sum_j^N \hat{w}_i^T \hat{w}_j - \sum_i^N \| \hat{w} \|^2 = \text{Sum}(\hat{W}\hat{W}^T) - N$, where the $\text{Sum}(\cdot)$ operator calculates the sum of all elements in the matrix. Since $N$ is a constant, it suffices to consider the first term $\text{Sum}(\hat{W}\hat{W}^T)$ only.

Since $\hat{W}\hat{W}^T$ is a positive semi-definite matrix, all its eigenvalues are nonnegative. Since $\hat{w}_i$ is a normalized vector, every diagonal element of $\hat{W}\hat{W}^T$ is 1. Then the trace of $\hat{W}\hat{W}^T$, which is also the sum of the eigenvalues, equals to $N$.

Table 1: Experimental result on language modeling (perplexity). Bold numbers denote for the best result.

| Model | Parameters | Validation | Test |
|---|---|---|---|
| 2-layer skip connection LSTM (Mandt et al., 2017) (tied) | 24M | 69.1 | 65.9 |
| AWD-LSTM (Merity et al., 2018) (w.o. finetune) | 24M | 69.1 | 66.0 |
| AWD-LSTM (Merity et al., 2018) (w.t. finetune) | 24M | 68.6 | 65.8 |
| AWD-LSTM (Merity et al., 2018) + continuous cache pointer | 24M | 53.8 | 52.0 |
| MLE-CosReg (w.o. finetune) | 24M | 68.2 | 65.2 |
| MLE-CosReg (w.t. finetune) | 24M | 67.1 | 64.1 |
| MLE-CosReg + continuous cache pointer | 24M | **51.7** | **50.0** |

Table 2: Experimental results on WMT English $\rightarrow$ German and German $\rightarrow$ English translation. Bold numbers denote for our results and ‡ denotes for our implementation.

| English→German | | German→English | |
|---|---|---|---|
| **Model** | **BLEU** | **Model** | **BLEU** |
| ConvS2S (Gehring et al., 2017) | 25.16 | DSL (Xia et al., 2017b) | 20.81 |
| `Base` Transformer (Vaswani et al., 2017) | 27.30 | Dual-NMT (Xia et al., 2017a) | 22.14 |
| `Base` Transformer + MLE-CosReg | **28.38** | ConvS2S‡ (Gehring et al., 2017) | 29.61 |
| `Big` Transformer (Vaswani et al., 2017) | 28.40 | `Base` Transformer‡ (Vaswani et al., 2017) | 31.00 |
| `Big` Transformer + MLE-CosReg | **28.94** | `Base` Transformer + MLE-CosReg | **31.93** |

As the cosine similarities between the word embeddings are all positive. According to Theorem 4, when $\hat{W}\hat{W}^T$ is a positive matrix, we have the largest absolute eigenvalue of $\hat{W}\hat{W}^T$ is upper bounded by $\text{Sum}(\hat{W}\hat{W}^T)$. Then minimizing $R$ is equivalent to minimize the upper bound of the largest eigenvalue. As the sum of all eigenvalues is a constant, the side effect is to increase other eigenvalues, consequently improving the expressiveness of the embedding matrix.

**Theorem 4.** *(Merikoski, 1984) For any matrix A, which is a real and nonnegative $n \times n$ matrix. The spectral radius, which is the largest absolute value of A's eigenvalues, is less than or equals to $Sum(A)$.*

## 6 EXPERIMENTS

We conduct experiments on two basic natural language generation tasks: language modeling and machine translation, and report the results in this section.

### 6.1 EXPERIMENTAL SETTINGS

#### 6.1.1 LANGUAGE MODELING

Language modeling is one of the fundamental tasks in natural language processing. The goal is to predict the probability of the next word conditioned on previous words. The evaluation metric is perplexity. Smaller the perplexity, better the performance. We used WikiText-2 (WT2) corpus, which is popularly used in many previous works (Merity et al., 2017; Inan et al., 2017; Grave et al., 2017). WikiText-2 is sourced from curated Wikipedia articles and contains approximately a vocabulary of over 30,000 words. All the text has been tokenized and processed with the Moses tokenizer (Koehn et al., 2006). Capitalization, punctuation and numbers are retained in this dataset.

AWD-LSTM (Merity et al., 2018) is the state-of-the-art model for language modeling. We directly followed the experimental settings as in Merity et al. (2018) to set up the model architecture and

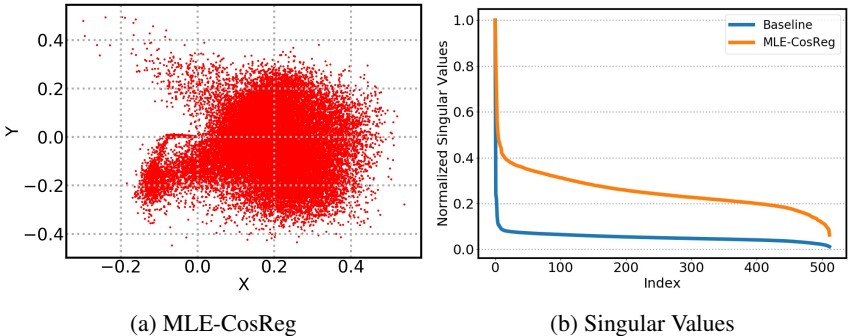

(a) MLE-CosReg  (b) Singular Values

Figure 2: (a): Word embeddings trained from MLE-CosReg. (b): Singular values of embedding matrix. We normalize the singular values of each matrix so that the largest one is 1.

hyperparameter configurations. We used a three-layer LSTM with 1150 units in the hidden layer and set the size of embedding to be 400. The ratio for dropout connection on recurrent weight is kept the same as Merity et al. (2018). We trained the model with Averaged Stochastic Gradient Descent. Our implementation was based on open-sourced code[3] by Merity et al. (2018). For our proposed MLE-CosReg loss, we found the hyperparameter $\gamma$ is not very sensitive and we set it to 1 in the experiments. We added neural cache model (Grave et al., 2017) to further reduce perplexity.

### 6.1.2 MACHINE TRANSLATION

Machine Translation aims at mapping sentences from a source domain to a target domain. We focus on English → German and German → English in our experiments. We used the dataset from standard WMT 2014, which consists of 4.5 million English-German sentence pairs and has been widely used as the benchmark for neural machine translation (Vaswani et al., 2017; Gehring et al., 2017). Sentences were encoded using byte-pair encoding (Sennrich et al., 2016), after which we got a shared source-target vocabulary with about 37000 subword units. We measured the performance with tokenized case-sensitive BLEU (Papineni et al., 2002).

We used state-of-the-art machine translation model Transformer (Vaswani et al., 2017), which utilizes self-attention mechanism for machine translation. We followed the setting in Vaswani et al. (2017) and used the official code (Vaswani et al., 2018) from Transformer. For both English → German and German → English tasks, we used the `base` version of Transformer (Vaswani et al., 2017), which has a 6-layer encoder and 6-layer decoder, the size of hidden nodes and embedding are set to 512. For English → German task, we additionally run an experiment on the `big` version of Transformer, which has 3x parameters compared with the `base` variant. All the models were trained with Adam optimizer, and all the hyperparameters were set as default as in Vaswani et al. (2017). $\gamma$ is set to 1 as in the experiments of language modeling.

### 6.2 EXPERIMENTAL RESULTS

We present experimental results for language modeling in Table 1 and machine translation in Table 2.

For language modeling, we compare our method with vanilla AWD-LSTM (Merity et al., 2018) in three different settings, without finetune, with finetune and with further continuous cache pointer. Our method outperforms it with 0.8/1.7/2.0 improvements in terms of test perplexity. For machine translation, comparing with original `base` Transformer (Vaswani et al., 2017), our method improves performance with 1.08/0.93 for the English → German and German → English tasks, respectively, and achieves 0.54 improvement on the `big` Transformer.

Note that for all tasks, we only add one regularization term to the loss function, while no additional parameters or architecture/hyperparameters modifications are applied. Therefore, the accuracy improvements purely come from our proposed method. This demonstrates that by regularizing the similarity between word embeddings, our proposed MLE-CosReg loss leads to better performance.

---

[3]https://github.com/salesforce/awd-lstm-lm

### 6.3 DISCUSSION

The study above demonstrates the effectiveness of our proposed method in terms of final accuracy. However, it is still unclear whether our method has improved the representation power of learnt word embeddings. In this subsection, we provide a comprehensive study on the expressiveness of the model learnt by our algorithm.

For a fair comparison with the empirical study in Section 3, we analyze the word embeddings of our model on English → German translation task. We project the word embeddings into 2-dimensional space using SVD for visualization. Figure 2(a) shows that the learnt word embeddings are somewhat uniformly distributed around the origin and not strictly in a narrow cone like in Figure 1(a). This shows that our proposed regularization term effectively expands word embedding space. We also compare the singular values for word embedding matrix of our learnt model and the baseline model, as shown in the Figure 2(b). According to the figure, trained with vanilla Transformer, only a few singular values dominate among all singular values, while trained with our proposed method, the singular values distribute more uniformly. Again, this demonstrates the diversity of the word embeddings learnt by our method.

## 7 CONCLUSION AND FUTURE WORK

In this work, we described and analyzed the *representation degeneration problem* in training neural network models for natural language generation tasks both empirically and theoretically. We proposed a novel regularization method to increase the representation power of word embeddings explicitly. Experiments on language modeling and machine translation demonstrated the effectiveness of our method.

In the future, we will apply our method to more language generation tasks. Our proposed regularization term is based on cosine similarity. There may exist some better regularization terms. Furthermore, it is interesting to combine with other approaches, e.g. (Gong et al., 2018), to enrich the representation of word embeddings.

## 8 ACKNOWLEDGEMENTS

This work was partially supported by National Basic Research Program of China (973 Program) (grant no. 2015CB352502), NSFC (61573026) and BJNSF (L172037). We would like to thank Zhuohan Li and Chengyue Gong for helpful discussions, and the anonymous reviewers for their valuable comments on our paper.

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

## APPENDICES

## A  PROOFS

In this section, we add proofs of all the theorems in the main sections.

**Theorem 1.** *A. If the set of uniformly negative direction is not empty, it is convex. B. If there exists a $v$ that is a uniformly negative direction of $h_i$, $i = 1, \cdots, M$, then the optimal solution of Eqn. 2 satisfies $\| w_N^* \| = \infty$ and can be achieved by setting $w_N^* = \lim_{k \to +\infty} k \cdot v$.*

*Proof.* The first part is straight forward and we just prove the second part. Denote $v$ as any uniformly negative direction, and $k$ as any positive value. We have $\lim_{k \to +\infty} \sum_{i=1}^{M} \log(\exp(\langle h_i, k \cdot v \rangle) + C_i) = \sum_{i=1}^{M} \log(C_i)$. As $\sum_{i=1}^{M} \log(C_i)$ is the lower bound of the objective function in Eqn. 2 and the objective function is convex, we have $w^* = \lim_{k \to \infty} k \cdot v$ is local optimum and also the global optimum. Note that the lower bound can be achieved only if $\langle h_i, w_N \rangle$ approaches negative infinity for all $i$. Then it is easy to check for any optimal solution $\| w_N^* \| = \infty$. □

**Theorem 2.** *There exists a $v$ that is a uniformly negative direction of a set of hidden states, if and only if the convex hull of the hidden states does not contain the origin.*

*Proof.* We first prove the necessary condition by contradiction. Suppose there are $M$ hidden states in the set. If the convex hull of $h_i$, $i = 1, \cdots, M$, contains the origin and there exists a uniformly negative direction (e.g. $v$). Then from the definition of convex hull, there exists $\alpha_i$, $i = 1, \cdots, M$, such that $\sum_{i}^{M} \alpha_i h_i = 0$, $\alpha_i \geq 0$ and $\sum_i \alpha_i = 1$. Multiplying $v$ on both sides, we have $\sum_i \alpha_i \langle h_i, v \rangle = 0$, which contradicts with $\langle h_i, v \rangle < 0$ for all $i$.

For the sufficient part, if the convex hull of $h_i$, $i = 1, \cdots, M$, does not contain the origin, there exists at least one hyperplane $H$ that passes through the origin and does not cross the convex hull. Then it is easy to check that a normal direction of the half space derived by the $H$ is a uniformly negative direction. The theorem follows. □

**Theorem 3.** *Given an $\alpha$-strongly convex function $f(x)$ and a function $g(x)$ that satisfies its Hessian matrix $\mathbf{H}(g(x)) \succ -\beta I$, where $I$ is the identity matrix, and $|g(x)| < B$. For a given $\epsilon > 0$, let $x^*$ and $x_\epsilon^*$ be the optimum of $f(x)$ and $(1 - \epsilon)f(x) + \epsilon g(x)$, respectively. If $\epsilon < \frac{\alpha}{\alpha+\beta}$, then $\| x^* - x_\epsilon^* \|_2^2 \leq \frac{4\epsilon B}{\alpha - \epsilon(\alpha+\beta)}$.*

*Proof.* We first prove the function $(1 - \epsilon)f(x) + \epsilon g(x)$ is $\alpha - \epsilon(\alpha + \beta)$-strongly convex.

Let's consider the Hessian matrix of it. As $f(x)$ is $\alpha$-strongly convex and $\mathbf{H}(g(x)) \succ -\beta I$, using the definition of positive-definite matrix, the following inequality holds:

$$\forall v, v^T (\mathbf{H}(g) + \beta I)v > 0; \tag{7}$$

$$v^T (\mathbf{H}(f) - \alpha I)v > 0. \tag{8}$$

To make it clear, we omit $x$ here. Then for the Hessian matrix of $(1 - \epsilon)f(x) + \epsilon g(x)$, we have:

$$\forall v, \quad v^T (\mathbf{H}((1 - \epsilon)f + \epsilon g) - (\alpha - \epsilon(\alpha + \beta))I)v \tag{9}$$

$$= v^T (\mathbf{H}((1 - \epsilon)f) + \mathbf{H}(\epsilon g) - (1 - \epsilon)\alpha I + \epsilon \beta I)v \tag{10}$$

$$= v^T ((1 - \epsilon)\mathbf{H}(f) + \epsilon \mathbf{H}(g) - (1 - \epsilon)\alpha I + \epsilon \beta I)v \tag{11}$$

$$= v^T ((1 - \epsilon)(\mathbf{H}(f) - \alpha I) + \epsilon(\mathbf{H}(g) + \epsilon \beta I))v \tag{12}$$

$$= v^T ((1 - \epsilon)(\mathbf{H}(f) - \alpha I))v + v^T (\epsilon(\mathbf{H}(g) + \beta I))v \tag{13}$$

$$= (1 - \epsilon)v^T (\mathbf{H}(f) - \alpha I)v + \epsilon v^T (\mathbf{H}(g) + \beta I)v \tag{14}$$

$$> (1 - \epsilon)0 + \epsilon 0 \tag{15}$$

$$= 0. \tag{16}$$

Thus, $\mathbf{H}((1-\epsilon)f(x)+\epsilon g(x))-(\alpha-\epsilon(\alpha+\beta))I$ is positive-definite, which means $(1-\epsilon)f(x)+\epsilon g(x)$ is $\alpha-\epsilon(\alpha+\beta)$-strongly convex. Then, with the properties in strong convexity, we have:

$$\| x^* - x_\epsilon^* \|_2^2 \quad \leq \quad \frac{2}{\alpha - \epsilon(\alpha + \beta)}(f(x^*) + \epsilon g(x^*) - f(x_\epsilon^*) - \epsilon g(x_\epsilon^*)) \tag{17}$$

$$\leq \quad \frac{2}{\alpha - \epsilon(\alpha + \beta)}(\epsilon g(x^*) - \epsilon g(x_\epsilon^*)) \tag{18}$$

$$\leq \quad \frac{4\epsilon B}{\alpha - \epsilon(\alpha + \beta)}. \tag{19}$$

$\square$

## B    COMPUTATION OF THE COSINE REGULARIZATION

In this section, we provide analysis of the computational cost of the proposed regularization term.

**Proposition 1.** *The cosine regularization in Eqn. 6 can be computed in $\Theta(N)$ time where $N$ is the size of vocabulary.*

*Proof.* The regularization term can be simplified as below:

$$\sum_i^N \sum_{j \neq i}^N \hat{w}_i^T \hat{w}_j \quad = \quad (\sum_i^N \hat{w}_i)^T (\sum_j^N \hat{w}_j) - \sum_i^N \hat{w}_i^T \hat{w}_i \tag{20}$$

$$= \quad (\sum_i^N \hat{w}_i)^T (\sum_i^N \hat{w}_i) - N \tag{21}$$

$$= \quad \| \sum_i^N \hat{w}_i \|_2^2 - N. \tag{22}$$

From the above equations, we can see that we only need to compute the sum of all the normalized word embedding vectors, and thus the computational time is linear with respect to the vocabulary size $N$. $\square$

## C    DISCUSSION ON LAYER NORMALIZATION

In this section, we show the condition on the existence of the uniformly negative direction holds almost for sure in practice with models where layer normalization is applied before last layer.

Let $h_1, h_2, \cdots, h_n$ be $n$ vectors in $\mathbb{R}^d$ Euclidean space. Let $\overrightarrow{\mathbf{1}} / \overrightarrow{\mathbf{0}}$ be the d-dimensional vector filled with ones/zeros respectively. By applying layer normalization we have:

$$\mu_i \quad = \quad \frac{1}{d}\overrightarrow{\mathbf{1}}^T h_i, \tag{23}$$

$$\sigma_i^2 \quad = \quad \frac{1}{d} \| h_i - \overrightarrow{\mathbf{1}}\mu_i \|_2^2, \tag{24}$$

$$h_i' \quad = \quad \mathbf{g} \odot \frac{h_i - \overrightarrow{\mathbf{1}}\mu_i}{\sigma_i} + \mathbf{b}, \tag{25}$$

where $\mu_i$ and $\sigma_i^2$ are the mean and variance of entries in vector $h_i$, $\mathbf{g}$ and $\mathbf{b}$ are learnt scale/bias vectors and $\odot$ denotes the element-wise multiplication. Here we simply assume that $\sigma_i^2$ and each entry in $\mathbf{g}$ and $\mathbf{b}$ are not zero (since the exact zero value can be hardly observed using gradient optimization in real vector space). If the convex hull of $h_1', h_2', \cdots, h_n'$ contains the origin, then there exist $\lambda_1, \lambda_2, \cdots, \lambda_n$, such that:

$$\sum_{i=1}^n \lambda_i h_i' = \overrightarrow{\mathbf{0}}; \sum_{i=1}^n \lambda_i = 1; \lambda_i \geq 0, \forall i = 1, 2, \cdots, n. \tag{26}$$

By combining 25 and 26 we have:

$$\sum_{i=1}^{n} \lambda_i h_i' \quad = \quad \sum_{i=1}^{n} \lambda_i (\mathbf{g} \odot \frac{h_i - \overrightarrow{\mathbf{1}} \mu_i}{\sigma_i} + \mathbf{b}) \tag{27}$$

$$= \quad \mathbf{b} + \sum_{i=1}^{n} \lambda_i (\mathbf{g} \odot \frac{h_i - \overrightarrow{\mathbf{1}} \mu_i}{\sigma_i}) \tag{28}$$

$$= \quad \mathbf{b} + \mathbf{g} \odot \sum_{i=1}^{n} \lambda_i \frac{h_i - \overrightarrow{\mathbf{1}} \mu_i}{\sigma_i} = \overrightarrow{\mathbf{0}}. \tag{29}$$

Denote $\frac{\mathbf{b}}{\mathbf{g}} = (\frac{b_1}{g_1}, \frac{b_2}{g_2}, \cdots, \frac{b_d}{g_d})$, thus we have $\sum_{i=1}^{n} \lambda_i \frac{h_i - \overrightarrow{\mathbf{1}} \mu_i}{\sigma_i} = -\frac{\mathbf{b}}{\mathbf{g}}$. Since $\overrightarrow{\mathbf{1}}^T \frac{h_i - \overrightarrow{\mathbf{1}} \mu_i}{\sigma_i} = 0$ for all $i$, there exist $\lambda_1, \lambda_2, \cdots, \lambda_n$ that satisfy Eqn. 26 only if $\overrightarrow{\mathbf{1}}^T \frac{\mathbf{b}}{\mathbf{g}} = -\overrightarrow{\mathbf{1}}^T \sum_{i=1}^{n} \lambda_i \frac{h_i - \overrightarrow{\mathbf{1}} \mu_i}{\sigma_i} = -\sum_{i=1}^{n} \lambda_i \overrightarrow{\mathbf{1}}^T \frac{h_i - \overrightarrow{\mathbf{1}} \mu_i}{\sigma_i} = 0$, which can hardly be guaranteed using current unconstrained optimization. We also empirically verify this.

## D  (SUB)WORD FREQUENCY DISTRIBUTION ON WMT2014 ENGLISH-GERMAN AND WIKITEXT 2 DATASETS

In this section, we show the (sub)word frequency distribution on the datasets we use in experiments in the Figure 3.

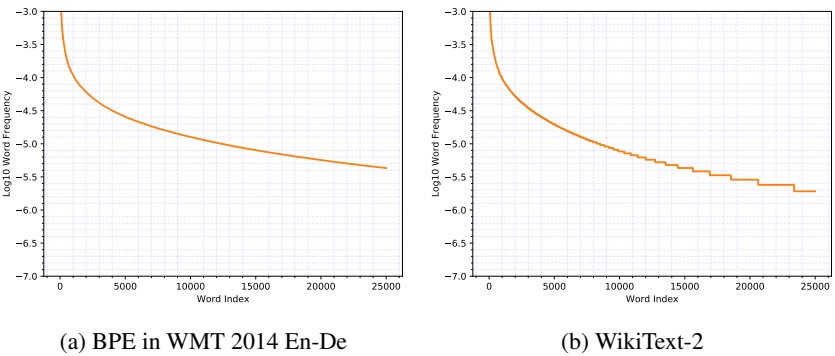

(a) BPE in WMT 2014 En-De

(b) WikiText-2

Figure 3: (a): WMT 2014 English-German Dataset preprocessed with BPE. (b): word-level WikiText-2. In the two figures, the x-axis is the token ranked with respect to its frequency in descending order. The y-axis is the logarithmic value of the token frequency.

