# OpenReview forum: "Representation Degeneration Problem in Training Natural Language Generation Models"
_ICLR.cc/2019/Conference_

### Official Review · AnonReviewer2 · 2018-10-31

**Rating:** 7
**Confidence:** 4

**Review:**

The paper presents and discusses a new phenomenon that infrequent words tend to learn degenerate embeddings. A cosine regularization term is proposed to address this issue.

Pros
1. The degenerate embedding problem is novel and interesting.
2. Some positive empirical results.

Cons and questions
1. The theory in Section 4 suggests that the degeneration problem originates from underfitting; i.e., there's not enough data to fit the embeddings of the infrequent words, when epsilon is small. However, the solution in Section 5 is based on a regularization term. This seems contradictory to me because adding regularization to an underfit model would not make it better. In other words, if there's not enough data to fit the word embeddings, one should feed more data. It seems that a cosine regularization term could only make the embeddings different from each other, but not better.
2. Since this is an underfitting problem (as described in Section 4), I'm wondering what would happen on larger datasets. The claims in the paper could be better substantiated if there are results on larger datasets like WT103 for LM and en-fr for MT. Intuitively, by increasing the amount of total data, the same word gets more data to fit, and thus epsilon gets large enough so that degeneration might not happen.
3. "Discussion on whether the condition happens in real practice" below Theorem 2 seems not correct to me. Even when layer normalization is employed and bias is not zero, the convex hull can still contain the origin as long as the length of the bias vector is less than 1. In fact, this condition seems fairly strong, and surely it will not hold "almost for sure in practice".
4. The cosine regularization term seems expensive, especially when the vocab size is large. Any results in terms of computational costs? Did you employ tricks to speed it up?
5. What would happen if we only apply the cosine term on infrequent words? An ablation study might make it clear why it improves performance.

UPDATE:
I think the rebuttal addresses some of my concerns. I am especially glad to see improvement on en-fr, too. Thus I raised my score from 5 to 7.

---

> ### Author Response · Authors · 2018-11-20
> **Rebuttals from authors [additional results on WMT 2014 En-Fr]**
>
> We thanks the reviewer for the comments.
>
> Q1. The theory in Section 4 suggests that the degeneration problem originates from underfitting, and should be solved by feeding more data instead of regularization.
>
> ANSWER:
> Feeding more data cannot fix the problem here. In natural language, it is known that no matter how large the dataset is, the frequency of any word is likely to be inversely proportional to its rank in the frequency table (referred as Zipf’s law), and thus the appearance of a large number of rare words in the dataset at any scale is inevitable. To show this, we provide statistics in the appendix on the WMT 2014 EnDe dataset, which has 4.5M sentence pairs and 261M tokens after BPE preprocessing. It can be seen from Figure 3, the number of rare tokens is large, while their frequencies are relatively small, which justifies that the problem remains even in a very large dataset. We have tried simple approach such as upsampling rare tokens and downsampling popular tokens to balance the expected loss per different words but it didn’t work well.
>
> Second, from our empirical observation and theoretical justification, we find that most tokens, especially rare appeared tokens are likely to be clustered together in the embedding space.  Unlike standard regularization to restrict the parameter space, our solution to this is direct **increasing** the distance between each embedding pairs. We find it is very useful to solve the problem and we are willing to change the term **regularization** to others if needed.
>
>
> Q2. Experiments on larger datasets
>
> ANSWER:
> Due to time limitation, we just completed experiments on the WMT En-Fr dataset during this rebuttal period, we achieved 43.29 BLEU score in Transformer-big model on the task, which is also better than the baseline. Together with the tasks completed in our paper, we think the method we propose is convincing to improve the models in different tasks.
>
> Q3. "Discussion on whether the condition happens in real practice"
>
> ANSWER:
> Thanks for pointing this out. We find that we have made a mistake but it doesn’t hurt the conclusion. We have revised the related paragraph and provided a formal analysis of how layer normalization affects the space of hidden states in Appendix (Page 12)
>
> Q4. The cosine regularization term seems expensive.
>
> ANSWER:
> The proposed regularizer can be computed in linear time with respect to the vocabulary size. We provide this mathematical simplification in Appendix (Page 12).
>
> Q5:  How about applying cosine regularization to rare words only.
>
> ANSWER:
> From Figure 3, we can see that the number of rare tokens (e.g., relative frequency < 10^{-4}) is still huge, so there is little experimental difference between applying the proposed loss to the whole vocabulary and to the rare words only. Not mention that an additional parameter (threshold) is needed to define what is **rare**.

---

> > ### Author Response · Authors · 2018-12-07
> > **Thanks for your attention.**
> >
> > Dear reviewer, we believe we have addressed your concerns and clarified your points in the rebuttal. Do you have an updated assessment (or concerns) of our paper? Thanks for your consideration.

---

> > > ### Comment · AnonReviewer2 · 2018-12-08
> > > **update**
> > >
> > > I just updated my scores. Thanks for your clarification and update.

---

### Official Review · AnonReviewer1 · 2018-11-01
**A simple regularization to solve a representation degeneration problem**

**Rating:** 7
**Confidence:** 3

**Review:**

This work proposes a simple regularization term which penalize cosine similarity of word embedding parameters in the loss function. The motivation comes from empirical studies of word embedding parameters in three tasks, translation, word2vec and classification, and showed that the parameters for the translation task are not distributed when compared with other tasks. The problem is hypothesized by the rare word problem especially when parameters are tied for softmax and input embedding, and proposes a cosine similarity regularization. Experiments on English/German show consistent gains over non-regularized loss.

Pros:

-  The proposed method is well motivated from empirical studies by visualizing parameters of three tasks, and the analysis on rare words are convincing.

- Good performance in language modeling and translation tasks by incorporating the proposed regularization.

Cons:

- The visualization might be slightly miss leading in that the size of classification, e.g., the vocabulary size, is different, e.g., BPE for translation, word for word2vec and categories of MNIST. I'd also like to see visualization for comparable experiments, e.g., language modeling with or without tied parameters.

- Given that BPE is used in translation, the analysis might not hold since rare words would not occur very frequently, and thus, the gain might come from other factors, e.g., tied source/target embedding parameters in Transformer.

- I'd like to see experiments under un-tided parameters with the proposed regularization.

---

> ### Author Response · Authors · 2018-11-17
> **Rebuttal from authors**
>
> We thank the reviewer for the insightful comments.
>
> Q1: On un-tied parameters and experiment.
>
> ANSWER:
> Our contribution is to understand the word embedding in language generation task with weight tying trick which is commonly used in the state-of-the-art models. All of our empirical/theoretical analysis are based on this setting by using the **tying** property. Therefore it is not that reasonable to extend our theories or apply our loss function to the un-tied setting and make comparisons.
>
> For example, it is easy to realize that without the weight tying trick, the embedding of rare words will be rarely updated, and thus they are likely to be nearly perpendicular to each other and around the origin if the embeddings are independently initialized using Gaussian distribution with zero mean and a small std like 0.01. Such phenomena are completely different from what we observed under the weight tying trick and cannot motivate the same solution.
>
> In fact, using weight tying trick or not can be considered as two extreme settings. If we do not use the trick, the embeddings of rare words are rarely updated and are likely to be nearly perpendicular to each other, while if we use the trick, the rare embeddings are updated to be similar to each other according to our theories. Our solution can be considered as a way to mitigate the disadvantages of these two settings.
>
>
> Q2: Regarding word token and sub word tokens (BPE).
>
> ANSWER:
> For translation tasks, we use sub-word tokens. However, according to our study, the sub-word frequency distribution is similar to the word level one (the statistics and figures are provided in the appendix). From Figure 3 in the appendix, we can see that with BPE, there still exists a large number of rare subwords in the training data.  Our experiments also show that by improving the expressiveness of the embeddings for tasks with either BPE-level tokens or word-level tokens, we achieve similar gain over the baselines.

---

### Official Review · AnonReviewer3 · 2018-11-02
**A new understanding of word embedding in LM and NMT**

**Rating:** 7
**Confidence:** 3

**Review:**

The authors propose a new understanding of word embedding in natural language generation tasks like language model and neural machine translation.
The paper is clear and original. The experiment results support their argument.

The problem they raised is quite interesting, however, it is not clear why the representation degeneration problem is important in language generation performance. In Figure 1, the classification is from MNIST, which is much different from words. The authors might want to explain more clearly why the uniformly distributed singular values are helpful in language generation tasks.

---

> ### Author Response · Authors · 2018-11-17
> **Rebuttal from authors**
>
> We thank the reviewer for the positive feedback.
>
> Q: why the representation degeneration problem is important in language generation
>
> ANSWER:
>
> We did make some discussions regarding the problem in the second paragraph of the intro section and section 3.2, we clarify it here:
>
> In the language generation tasks, the word embedding parameters are tied with softmax weight matrix in the last layer, and thus it has a dual role in the model, serving as the input in the first layer and the weights in the last layer. The representation degeneration problem is important from the below two aspects:
>
> (1). Given its first role as input word embedding, it should be widely distributed to represent different semantic meanings which will be further used for different tasks. However, we observe that most of the trained word embedding in language generation tasks are positively correlated and spread in a narrow cone, which limits the expressiveness of the semantic word representations.
> (2). Given its role as output softmax matrix, to achieve good prediction of next word in a target sentence, a more diverse distribution of word embeddings in the space is expected to obtain a large margin result with good generalization.
>
> According to the discussion above, we think the current learnt model needs improving.
>
> We are not exactly targeting to have **a more uniform spectral density distribution** but there are some works which show that more uniformly distributed singular values of embedding matrix can bring better performance. [1] shows that by using simple post-processing approaches (removing the first several principal components in learnt word embeddings), we can get better performance of several downstream classification tasks.
>
> [1]. Jiaqi Mu, Suma Bhat, and Pramod Viswanath. All-but-the-top: simple and effective postprocessing for word representations. ICLR 2018.

---

### Meta-Review · Area_Chair1 · 2018-12-14
**limited contribution but well executed**

**Confidence:** 3
**Recommendation:** Accept (Poster)

**Metareview:**

although i (ac) believe the contribution is fairly limited (e.g., (1) only looking at the word embedding which goes through many nonlinear layers, in which case it's not even clear whether how word vectors are distributed matters much, (2) only considering the case of tied embeddings, which is not necessarily the most common setting, ...), all the reviewers found the execution of the submission (motivation, analysis and experimentation) to be done well, and i'll go with the reviewers' opinion.